# DOES MIM CHEAT? MEASURING SEMANTIC INVARIANCE IN SELF-SUPERVISED ViT REPRESENTATIONS

## ABSTRACT

In this work, we analyze patch-level embeddings and show that MIM objectives bias representations toward non-semantic cues, limiting their effectiveness in inference. To probe this effect, we introduce a model-agnostic counterfactual score that quantifies *semantic invariance* by comparing principal component responses to real inputs and noise, providing a novel measure to directly characterize the tradeoff between semantic information and structural noise in ViT embeddings. Building on this measure, we propose Semantic Orthogonal Projection (SOaP), a post-hoc method using simple Gram–Schmidt orthogonalization that suppresses invariant components in patch representations. Our experiments show that SOaP consistently improves performance on multiple downstream tasks across state-of-the-art MIM-based models.

## 1 INTRODUCTION

Self-supervised learning (SSL) has become a dominant approach for training vision transformers (ViTs), yielding models that generalize well across diverse downstream tasks (Zhou et al., 2022; Caron et al., 2021; Oquab et al., 2024; Siméoni et al., 2025; Darcet et al., 2025; Tschannen et al., 2025; Venkataramanan et al., 2025). Current approaches fall broadly into two categories: contrastive

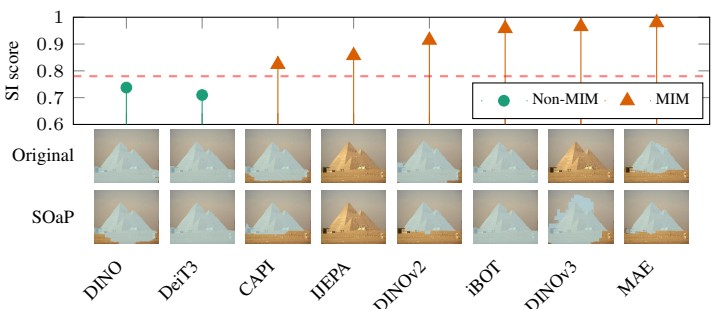

**Figure 1: Top:** Models with MIM objectives exhibit higher semantic invariance (SI) scores than models without MIM objectives. **Bottom:** We show on-par or improved zero-shot salient segmentation results between original patch embeddings (first row) and our proposed SOaP (second row), both evaluated using a few leading components.

objectives, which learn instance-level invariances to input augmentations (Caron et al., 2021), and masked image modeling (MIM), which trains models to reconstruct masked patches from surrounding context (Assran et al., 2023; Darcet et al., 2025; He et al., 2022; Baevski et al., 2022; Alkin et al., 2025). The contrastive objectives act on the instance-level global representations, and can be seen as an instance classification task (Chen et al., 2023). Meanwhile, MIM objectives operate at the local patch representations, which can be reconstructed in the latent space (Zhou et al., 2022; Oquab et al., 2024; Darcet et al., 2025; Bar et al., 2024) or in the input space directly (He et al., 2022; Baevski et al., 2022; Alkin et al., 2025). Recent frameworks combine both these objectives (Oquab et al., 2024; Siméoni et al., 2025; Zhou et al., 2022), with contrastive losses shaping instance-level representations and MIM losses refining patch-level representations—together, the aim is to encourage models to capture both global semantics and fine-grained local structure.

Despite the success of MIM objectives, several works allude to high variance components and positional noise in the learned embeddings (Venkataramanan et al., 2025; Zhou et al., 2022; Oquab et al., 2024; Siméoni et al., 2025; Darcet et al., 2025; Przewięzlikowski et al., 2025). This is unsurprising, as the MIM task requires predicting both the semantic content and location of the masked patches (Bar et al., 2024). While positional collapse—where the model learns to predict the posi-

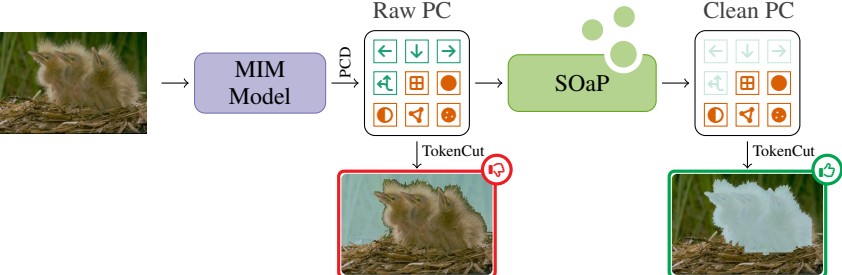

**Figure 2:** An image contains principal components (either semantic or non-semantic), extracted through principal component decomposition (PCD). We found that MIM-based models rank the non-semantic components higher than their semantic counterparts. Orthogonalizing the components with the highest semantic invariance (through SOaP) boosts performance in various downstream tasks—here we show zero-shot salient segmentation with TokenCut.

tion of masked tokens instead of content—is a known issue with various suggested work-arounds (Darcet et al., 2025; Bar et al., 2024; Venkataramanan et al., 2025), few studies directly address this phenomenon.

In this paper, we take a closer look at *structural noise* in ViT tokens for state-of-the-art MIM and contrastive models. We characterize structural noise as components that are invariant to the *semantic signal* in the input. This includes positional information, which remains present even after removing the semantic contents of an input. Identifying structural noise can then be posed as a comparative task between semantic and non-semantic images. Using principal component analysis of the representation spaces learned by modern SSL models, we discover that several leading components exhibit high levels of structural noise. Upon further inspection, we find that that several MIM-based models encode strong positional information, while models trained without MIM do not exhibit this behavior—see Figure 1. This observation holds true regardless of which positional embedding method is employed (Siméoni et al., 2025; Heo et al., 2024) and whether the MIM predictions are in the latent or input space, suggesting that this is a broader problem in MIM. The *consequence* of structural noise in the representations is an associated *trade-off with semantic information*, and *reduced performance* in dense prediction tasks, particularly in zero-shot settings (Venkataramanan et al., 2025; Vanyan et al., 2024). Therefore, our analysis also serves as a diagnostic tool to explore the trade-off between semantic and structural information in SSL representaitons.

Finally, taking all our observations into account, we introduce the Semantic Orthogonal Projection (SOaP), a novel off-the shelf denoising strategy that suppresses structural noise in pretrained SSL models, leading to improved representations for downstream—cf. Figure 2. SOaP is flexible. It is computed directly from data using a Gram-Schmidt based projection, thus requiring no training, and can be attached as an external module to any pretrained SSL backbone. In summary, our **contributions** include: (i) The interesting discovery that MIM objectives *uniquely* bias representations toward encoding structural noise rather than semantic information. (ii) A novel *Semantic Invariance Score* to measure the level of semantic invariance in a model's representations, allowing us to diagnose the semantics-structural noise trade-off in SSL representations. (iii) Finally, based on this score, we propose Semantic Orthogonal Projection (SOaP), a post-hoc denoising strategy that suppresses structural noise, and are able to demonstrate improved performance in zero-shot downstream tasks for all MIM-based models.

## 2 RELATED WORK

**Contrastive Learning.** The contrastive approach to self-supervised learning, wherein representations are encouraged to be invariant to view augmentations, has a extensive history (Chen et al., 2020a; Tian et al., 2020; Chen et al., 2020b; Wang & Isola, 2020; Zhang & Ma, 2022). More modern approaches adopt a knowledge distillation approach by matching representations between a teacher and a student network, with periodic exponential moving average (EMA) updates (Grill et al., 2020; Caron et al., 2020; Chen & He, 2021; Caron et al., 2021). While sometimes characterized as joint-embedding or matching-based methods, these approaches rely on strong global cues that match local with global views via aggressive cropping without preserving relative spatial information. However,

a stronger reliance on global cues limits performance on dense prediction tasks, as finer-grained semantic relationships are not exploited (Liu & Gould, 2024; Yuan et al., 2023; Wang et al., 2022; Puy et al., 2024).

**Masked Image Modeling.** A strong alternative to learning invariances through data augmentations, is through choosing masking as the pretext task (He et al., 2022; Xie et al., 2022; Bao et al., 2022). While contrastive learning is prone to model collapse (the trivial solution), MIM provides a more robust and scalable approach by directly reconstructing the input data. Though masking strategies vary, MAE (He et al., 2022) popularized dropping most patches, and reconstructs in the pixel space through a decoder. Reconstruction in the representation space, rather than the pixel space, circumvents the need for a heavy decoder, and has been shown to work in methods like I-JEPA (Assran et al., 2023). The MIM objective has been adopted in the self-distillation setup quite successfully— iBOT (Zhou et al., 2022) employs the averaged cross entropy over the student's masked view, and the teacher the non-masked view, and DINOv2 demonstrates additional improvements by separating the MLP projection heads, using adaptive resolutions, and additional regularization (Oquab et al., 2024). CAPI (Darcet et al., 2025) adopts a deep clustering, pure MIM approach by generating pseudo-labels in the masked latent space, thus circumventing multiple issues with the contrastive approaches relating to training instability, and sensitivity to hyperparameters. Most recently, DINOv3 improves over DINOv2 by using more modern positional embeddings, and a denoised warmup training phase (Siméoni et al., 2025). We consider all these approaches in our work, and our observations are consistent across these models.

**Positional Noise in MIM Models.** A recurring challenge in masked image modeling (MIM) frameworks is the presence of positional noise in the learned embeddings. This has been indirectly addressed through a variety of architectural and training modifications, such as introducing regularization mechanisms (Darcet et al., 2025; Siméoni et al., 2025), or directly addressed by disentangling positional cues (Venkataramanan et al., 2025; Yang et al., 2024; Bar et al., 2024; Wang et al., 2024). Despite these efforts, MIM representations typically show weaker out-of-the-box performance compared to frameworks with contrastive objectives, and often require extensive fine-tuning to transfer effectively to downstream tasks. Some works analyze embedding variance to identify noisy or unstable tokens (Vanyan et al., 2024), while others propose selective aggregation to prune non-informative dimensions (Przewięzlikowski et al., 2025). Venkataramanan et al. (2025) introduce RASA, a post-hoc module that suppresses explicit location cues by training to predict patch positions from pretrained embeddings. However, RASA only addresses patch-location noise and does not provide a broader analysis of structural noise in other frameworks. In contrast, our proposed solution requires no training. Additionally, our study goes beyond positional cues, probing the full spectrum of semantically invariant noise encoded by state-of-the-art SSL models, and proposing metrics to quantify and interpret these effects.

# 3 SEMANTIC ORTHOGONAL PROJECTION

Before we introduce our proposal, we provide an exploratory analysis that sheds insights to decompose the semantic and structural information from the data. Specifically, we use principal component analysis (PCA) to expose the structure that reveals a positional bias in the MIM training (Section 3.1). Then, we present a counterfactual strategy that identifies semantic invariant components by contrasting real and synthetic inputs (Section 3.2). Based on these observations, we propose a score to measure structural noise through semantic invariance (Section 3.3). Finally, we introduce our Semantic Orthogonal Projector (SOaP) which uses this score to suppress non-informative structural information, thus, enhancing future downstream tasks.

## 3.1 PRINCIPAL COMPONENT ANALYSIS OF PATCH EMBEDDINGS

To substantiate our view of embeddings as mixtures of signals, we analyze them using PCA. PCA decomposes the representation space into dominant variance directions, which allows us to examine whether leading components reflect semantic content or non-semantic structure, such as positional bias. We estimate the covariance of a set of patch embeddings $z \in \mathcal{Z}, z \in \mathbb{R}^D$ under a given model $f$ over ImageNet (Deng et al., 2009) using Welford's online algorithm (Welford, 1962), and obtain

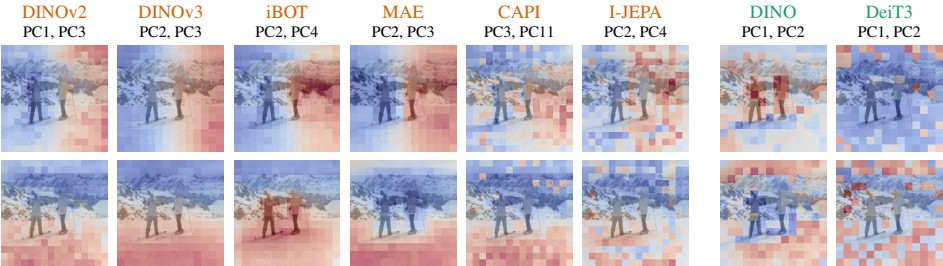

**Figure 3:** Patch representations from MIM models exhibit strong positional bias in leading (high rank) principal components, illustrated here for one example image. The vertical line separates MIM-based models from non-MIM based models. Models trained with MIM show a clear left/right and top/bottom bias in its leading principal components (PC1-PC11) as opposed to models trained without MIM.

the eigendecomposition

$$\text{Cov}(\mathcal{Z}) = V\Lambda V^\top \in \mathbb{R}^{D\times D}, \tag{1}$$

with principal component vectors $V = (v_1, \ldots, v_D)$. For each component $d$, we define the response of an input $z$ by the inner product $\langle z, v_d \rangle$, which reflects how strongly each patch embedding $z \in \mathcal{Z}$ projects onto direction $v_d$.

Applying this criterion, we observe that patch representations from MIM models exhibit strong positional bias in leading or highly ranked principal components, as illustrated in Figure 3. This indicates that a significant share of their variance is devoted to encoding position, implying that positional bias arises as a direct consequence of latent MIM objectives. In contrast, models trained without MIM objectives, such as DINO and DeiT3, do not exhibit this behavior.

To further characterize these components, we evaluate their behavior on a set of images. Let $\Omega$ be the input set of images, and $f : \Omega \to \mathcal{Z}$ a model that encodes an image $x_b \in \Omega$ into $N$ patch embedding representations $\boldsymbol{z}^{(b)} = \{z_n^{(b)}\}_{n=1}^N$. The binary activation of component $d$ for $x_b$ is then defined by thresholding the token responses

$$A_d(x_b) = \mathbf{1}[\boldsymbol{z}^{(b)}v_d \geq \tau] \in \{0,1\}^N. \tag{2}$$

Here, $\tau = \langle \nu, v_d \rangle$ is a scalar threshold, and $\nu$ is some threshold vector. In our experiments, we set $\nu = \mathbf{0}$. Averaging over images, $\bar{A}_d = \frac{1}{B}\sum_b A_d(x_b) \in \mathbb{R}^N$, yields an aggregated activation map that highlights spatial patterns.

With these definitions in place, we formulate two diagnostic conditions to identify non-semantic components. First, when the average activation $\bar{A}_d$ exhibits systematic dependence on patch location, we infer that component $d$ predominantly encodes positional cues rather than semantic content. Second, when the per-sample activations $A_d(x_b)$ exhibit minimal variation across different inputs, component $d$ can be regarded as invariant to image content, which indicates that it captures non-informative signals.

### 3.2 PATCH EMBEDDING AS LINEAR COMBINATIONS

To better isolate the signals underlying a model's representations, we express each patch embedding $z$ as a sum of semantic and non-semantic factors. Ideally, we want the model $f$ to capture the relevant sematic information in the input, like global class information and local structures of color, shape, texture, and fine-grained semantics useful for dense classification tasks such as semantic segmentation. However, specific to ViTs, $f$ must also encode the positional information directly to each embedding $z$, since the attention operator is otherwise permutation-invariant.

We formalize this by assuming that each embedding $z$ can be decomposed into two orthogonal subspaces, $\Phi$ for semantic information and $\mathrm{P}$ for structural information. Then we express

$$z = \theta_\phi \phi + \underbrace{\theta_\rho \rho + \varepsilon}_{\text{semantically non-informative}}; \quad \varepsilon \sim \mathbb{P}, \tag{3}$$

where $\theta_\phi, \theta_\rho \in \mathbb{R}_{\geq 0}$ are scalar coefficients for $\phi \in \Phi, \rho \in \mathrm{P}$, and $\varepsilon$ represents residual noise from some probability distribution $\mathbb{P}$. The semantic information component $\phi \in \Phi$ encodes both

relevant information for global objectives, such as instance discrimination and classification, and dense objectives like segmentation. The non-semantic information component $\rho \in \mathrm{P}$ encodes local positions and relative geometry among the local patch embeddings $z^{(b)}$, based on their positions in the original image $x_b$.[1]

Intuitively, both contrastive and MIM-based training objectives encourage $f$ to encode relevant semantic information. However, in line with the observations of previous work (Bar et al., 2024), we posit that the inpainting objective in MIM also drives $f$ to encode a substantial amount of non-semantic information $\rho$, thereby increasing the coefficient $\theta_\rho$ at the expense of the semantic coefficient $\theta_\phi$. This raises the question of how much of a learned representation reflects semantic content $\phi$, as opposed to structural noise or other non-informative signals. To address this, we introduce a *semantic invariance score* to measure the degree of non-semantic information encoded by each principal component $d$ in a learned patch representation space $\mathcal{Z}$.

### 3.3 SEMANTIC INVARIANCE

Semantic invariance (Yuan et al., 2024) refers to the property of a component yielding consistent responses even when the semantic content of local representations varies. In other words, a component is semantically invariant if it produces similar activations regardless of whether the input carries meaningful semantic information. Such components are uninformative for downstream understanding, as they do not encode or reflect actual image contents.

Let $\mathcal{X} \subset \Omega$ be the set of semantically informative images, and let $\mathcal{X}^c$ denote a complementary set without semantic information. In practice, $\mathcal{X}$ is instantiated as the ImageNet validation set (Deng et al., 2009), while $\mathcal{X}^c$ is approximated by a synthetic noise generator. The details of the generator are described in Appendix D—see the examples in Figure D.1. For each component $d$, we compute binary activations $A_d$ (2) for samples $x \sim \mathcal{X}$ and $x^c \sim \mathcal{X}^c$. This yields two empirical Bernoulli distributions of activations for the token index $n = 1, \dots, N$, such that

$$P_{d,n} = \Pr\big(A_{d,n} = 1 \mid x \sim \mathcal{X}\big); \quad Q_{d,n} = \Pr\big(A_{d,n} = 1 \mid x^c \sim \mathcal{X}^c\big). \tag{4}$$

If $P_{d,n} \approx Q_{d,n}$, then component $d$ behaves similarly for semantic and non-semantic inputs for token index $n$ and is therefore semantically invariant, hence non-informative. Conversely, large discrepancies between $Q_{d,n}$ and $Q_{d,n}$ indicate sensitivity to semantic content at position for index $n$, which posits $P_d, Q_d$ as multinomial distributions over all tokens. To quantify the discrepancy, we use a variant of the Dice-Sørensen coefficient (Carass et al., 2020)

$$s_d = \mathrm{SI}(P_d, Q_d) = 2 \cdot \frac{P_d \cdot Q_d + (1 - P_d) \cdot (1 - Q_d)}{\sqrt{P_d^2 + (1 - P_d)^2} + \sqrt{Q_d^2 + (1 - Q_d)^2}} = 2 \cdot \frac{\langle P_d, Q_d \rangle}{||P_d|| + ||Q_d||}. \tag{5}$$

A vectorial formulation of (5) is given in (C.1). This ensures that the score equals one *iff.* the two distributions coincide, while remaining robust when one distribution is highly skewed.

### 3.4 SEMANTIC ORTHOGONAL PROJECTOR (SOaP)

With SI score (5), we introduce Semantic Orthogonal Projector (SOaP), a novel off-the shelf denoising strategy that suppresses structural noise. We hypothesize that suppressing components that are invariant to semantic content in the representation $z$ acts as a denoising step, yielding representations that are closer to the semantic part $\theta_\phi \phi$ (3). To achieve this, we operate in the PCA basis, where these components $v_d$ are orthogonal by construction. Each component is assigned a weight $w_d$, derived from its semantic invariance score $s_d$ and a scaling function $t$, which determines how strongly it should be suppressed. Our SOaP projector $P_\phi$ is then defined following Gram–Schmidt based projection (Venkataramanan et al., 2025), subtracting the contribution of components identified as non-informative

$$P_\phi = I - VWV^\top; \quad \hat{z} = P_\phi z. \tag{6}$$

---

[1]We note that positional information is given implicitly in certain models that retain spatial order, like CNNs and MLPs. In this case $f$ does not need to explicitly encode positional information into $z$. However, we restrict this study to ViTs.

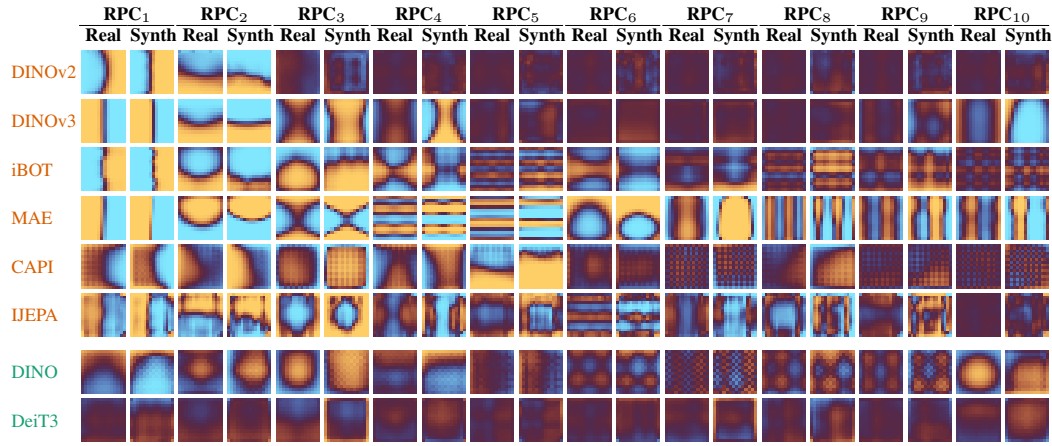

**Figure 4:** Distributions for the top 10 principal components, ranked by semantic invariance (RPC). **Real** columns correspond to $P_d$ and **Synth** correspond with $Q_d$ from Section 3.3, and each column compares the activations for real images from ImageNet, and generated synthetic images. MIM models are shows in the top rows; non-MIM models in the bottom. See appendix C for details.

Here, $V$ are the PCA components from (1), $I$ is the identity matrix, and $W = \text{diag}(w_1, \ldots, w_D)$ is a diagonal weight matrix. We propose a filter using a Fermi window (Bernstein et al., 2001; Caparelli & Tomasi, 2008)—commonly used in MRI imaging—using the rank of the scores $r$. This corresponds to a smooth regularization of the scores (Hansen, 2010) using sigmoid gating approach (Nguyen et al., 2024) with explicit control over truncation and smoothness. This formulation yields a principled weight estimation per component

$$w_d = t(s_d, r) = s_d \times \frac{\sigma((\mu - r)/\tau)}{\sigma(\mu/\tau)}; \quad r = \text{rank}(s_d). \tag{7}$$

Coupling statistical variance with semantic relevance, the hyperparameters $\mu$ and $\tau$ gives flexible control of the cut-off and smoothness of suppression, providing a principled way to adapt denoising strength to the spectral structure of embeddings. The result is a more stable projection operator that retains informative components while systematically attenuating noise, making the suppression sparse to focus on non-semantic components. Figure D.2 shows the effect of the scaling function on the semantic invariance scores.

## 4 EXPERIMENTS

For our study, we consider MAE (He et al., 2022), I-JEPA (Assran et al., 2023) and CAPI (Darcet et al., 2025) for models with MIM objectives, DINO (Caron et al., 2021) for global contrastive objective, and iBOT (Zhou et al., 2022), DINOv2 (Oquab et al., 2024), and DINOv3 (Siméoni et al., 2025) for models with both. For completeness we also evaluate supervised models RoPE-ViT (Heo et al., 2024) and DeiT-III (Touvron et al., 2022). Table 1 provides an overview of the models.

**Table 1:** Overview of models in our study. We select a representative group of models by including models with different architectures, objectives, MIM modes, and positional encoding in our experimental setup.

| Model | Arch. | Objective | MIM mode | Pos. enc. |
|---|---|---|---|---|
| DINOv2 | ViT-B/14 | MIM + CL | Latent | Add. |
| DINOv3 | ViT-B/16 | MIM + CL | Latent | RoPE |
| iBOT | ViT-B/16 | MIM + CL | Latent | Add. |
| MAE | ViT-B/16 | MIM | Pixel | Add. |
| CAPI | ViT-L/14 | MIM | Latent | Add. |
| I-JEPA | ViT-H/14 | MIM | Latent | Add. |
| DINO | ViT-B/16 | CL | NA | Add. |
| DeiT3 | ViT-B/16 | Supervised | NA | Add. |

### 4.1 ANALYZING INFORMATION CONTENT IN SSL TOKENS

We aggregate the activation maps of each component over patch embeddings from ImageNet validation and generated synthetic images—Figure 4 shows the top 10 principal components for each

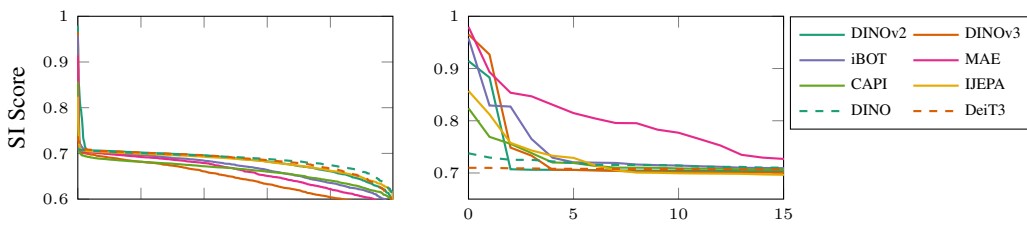

(a) Principal components ordered by score.    (b) Top 15 principal components.

**Figure 5:** Semantic invariance (SI) score in descending order. All principal components are shown in the left plot, while the right shows the top 15 semantically invariant scores up close. Note that all MIM-models have a max-score $\geq 0.75$, while all non MIM models have a lower score.

model, sorted by SI-scores. All MIM-based models reveal activations with strong positional bias in the form of top-bottom and left-right alignment. Critically, we do not observe similar top-bottom or left-right activations in DINO and DeiT3. This empirically supports the proposition that MIM strengthens the encoding of positional information in patch tokens. Furthermore, the phenomenon is present in both MIM-based models with standard additive positional embeddings (DINOv2, iBOT, CAPI, IJEPA) and positional embeddings injected by concatenation in the attention mechanism via RoPE (DINOv3). We also note that the two components with the highest semantic invariance are the exact same components identified as encoding positional noise in Figure 3 for all MIM models, showing that our SI-score captures positional and other sources of structural noise.

Next, we look at the SI-scores for the principal components of all the models—Figure 5 shows the scores in descending order. We observe that MIM-models exhibit much higher maximum SI-scores than models trained without MIM. In particular, at least two components have an SI-score higher than 0.75 for all MIM-models. Meanwhile, models trained without MIM score below this threshold for all components. Figure 1 shows the maximum SI-score for each model, highlighting the presence of semantically invariant principal components in MIM-models. We also probe semantic invariance over model depth. Figure 6 shows the maximum SI-score per layer for a selection of the models. We observe that semantic invariance starts out high in the early layers, which is expected for models with additive positional embeddings, although DeiT3 is a somewhat surprising exception to this. Specific to the MIM based models is that the SI-score increases again for the last layers. This can be explained as the model saturating more positional information into the embeddings in preparation for solving the MIM task. For MAE however, the score remains high accross all layers.

## 4.2 CLEANING WITH SOaP: EVALUATING DENSE REPRESENTATIONS

We use SOaP to correct for invariant components in local embeddings, and find that this improves performance in sparse and dense downstream tasks for all MIM models.

**Salient segmentation.**    We select TokenCut (Wang et al., 2023) for zero-shot evaluation of relative saliency information present in local embeddings. Table 2 shows that correcting the patch embeddings, in addition to using salient principal components as guides to foreground selection, can significantly boost performance.

We observe that DINO performs better out-of-the-box compared to the models with MIM objectives. We believe this is because DINO is trained with a global objective only. Positional information, and to a degree local semantic information, is not as useful for the salient segmentation task which relies on global (class specific) correlations. However, after suppressing semantically invariant components, most MIM models perform on par or better than DINO. Some notable exceptions are I-JEPA and DINOv3, which perform badly in general for this task.

**kNN classification.**    To probe the degree to which instance information is encoded in the embeddings for each model, we perform k-nearest neighbor classification on ImageNet (Deng et al., 2009). We compare the top-1 and top-5 accuracies of the average of the patch embeddings, and the corrected patch embeddings. The results in Table 3 show modest improvements after correcting for invariant components. We note that the benefit may be dampened by the aggregation, as spatial information about the locality of patches may cancel out by taking the average.

**Table 2:** Zero-shot salient segmentation with TokenCut. We evaluate on ECSSD (Yan et al., 2013), DUTS (Wang et al., 2017), and DUT-OMRON (Yang et al., 2013). Correcting the embeddings with SOaP improves results for all MIM-based models.

| Pretrain | Model | ECSSD | | | DUTS | | | DUT-OMRON | | |
|---|---|---|---|---|---|---|---|---|---|---|
| | | max $F_\beta$ | IoU | Acc. | max $F_\beta$ | IoU | Acc. | max $F_\beta$ | IoU | Acc. |
| *Original embeddings* | | | | | | | | | | |
| DINO | ViT-B16 | 82.580 | 74.325 | 90.929 | 75.769 | 89.705 | 83.932 | 52.851 | 83.019 | 59.289 |
| DINOv2 | ViT-B16 | 71.319 | 63.937 | 83.147 | 63.064 | 78.891 | 69.700 | 39.643 | 75.067 | 45.923 |
| DINOv3 | ViT-B16 | 36.975 | 29.122 | 52.953 | 31.302 | 52.264 | 39.874 | 15.623 | 46.258 | 19.656 |
| CAPI | ViT-L14 | 72.456 | 66.083 | 84.334 | 61.913 | 78.634 | 68.148 | 42.423 | 77.762 | 49.150 |
| iBOT | ViT-B16 | 62.873 | 56.248 | 78.785 | 59.657 | 77.987 | 66.353 | 29.602 | 67.883 | 33.731 |
| MAE | ViT-B16 | 79.952 | 71.067 | 89.410 | 70.227 | 86.078 | 79.229 | 45.630 | 78.552 | 54.758 |
| I-JEPA | ViT-H14 | 37.670 | 27.989 | 68.898 | 24.890 | 63.592 | 33.008 | 24.340 | 71.343 | 33.345 |
| *Corrected embeddings* | | | | | | | | | | |
| DINO | ViT-B16 | 81.017 | 70.416 | 90.237 | 73.178 | 85.333 | 80.627 | 44.004 | 68.709 | 49.136 |
| DINOv2 | ViT-B16 | 80.633 | 72.559 | 88.687 | 76.785 | 89.440 | 84.387 | 43.472 | 71.762 | 50.610 |
| DINOv3 | ViT-B16 | 42.633 | 33.742 | 61.975 | 39.057 | 63.033 | 47.624 | 17.485 | 51.390 | 23.329 |
| CAPI | ViT-L14 | **85.219** | **78.084** | **92.600** | **78.439** | **91.906** | **85.710** | **51.315** | **80.291** | **59.872** |
| iBOT | ViT-B16 | 66.557 | 60.167 | 78.340 | 65.618 | 80.595 | 72.192 | 31.991 | 63.552 | 36.330 |
| MAE | ViT-B16 | 82.094 | 72.118 | 91.444 | 72.293 | 90.107 | 82.877 | 48.297 | 82.974 | 59.931 |
| I-JEPA | ViT-H14 | 40.239 | 31.162 | 71.406 | 25.922 | 65.001 | 33.371 | 27.038 | 76.841 | 35.472 |

**Table 3:** kNN classification of average pooled patch embeddings on ImageNet (Deng et al., 2009). We compare the original embeddings with the SOaP-corrected embeddings for each backbone, and report top-1 and top-5 accuracies.

| Pretrain | Model | Original embeddings | | Corrected embeddings | |
|---|---|---|---|---|---|
| | | kNN Acc@1 | kNN Acc@5 | kNN Acc@1 | kNN Acc@5 |
| DINOv2 | ViT-B16 | 77.064 | 91.624 | 77.100 ↑ 0.036 | 91.636 ↑ 0.012 |
| DINOv3 | ViT-B16 | 76.542 | 91.530 | 76.588 ↑ 0.046 | 91.612 ↑ 0.082 |
| CAPI | ViT-L14 | 56.250 | 77.490 | 56.444 ↑ 0.194 | 77.742 ↑ 0.252 |
| iBOT | ViT-B16 | 59.170 | 79.612 | 59.498 ↑ 0.328 | 79.918 ↑ 0.306 |
| MAE | ViT-B16 | 47.488 | 69.168 | 47.758 ↑ 0.27 | 69.442 ↑ 0.274 |
| I-JEPA | ViT-H14 | 71.382 | 86.144 | 71.390 ↑ 0.008 | 86.168 ↑ 0.024 |
| DINO | ViT-B16 | 55.216 | 75.740 | 55.586 ↑ 0.370 | 75.900 ↑ 0.160 |
| DeiT3 | ViT-B16 | 79.914 | 91.820 | 79.956 ↑ 0.042 | 91.726 ↓ -0.094 |

### 4.3 ABLATIONS

**SI-score sensitivity to dataset choice.** To probe the SI-score sensitivity to dataset choice, we ablate over Caltech256 (Griffin et al., 2022), COCO-Stuff164k (Lin et al., 2014), CUB200 (Welinder et al., 2010), PASCAL VOC (Everingham et al., 2012), and ImageNet (Deng et al., 2009) using DINOv2 as the backbone. We calculate the cosine distance between the score vectors over the $D = 768$ components. Figure 7 shows the SI-score distance for each pair of datasets. We observe that the distances are low (between 0.0025 and 0.0032), which means that the scores for each component remain consistent despite using different datasets with different distributions and of various sizes to instantiate semantically informative input. This indicates that the SI-score is not sensitive to dataset choice.

**Comparison with RASA.** We compare SOaP with RASA by correcting the patch embeddings from Franca using pretrained RASA weights from Venkataramanan et al.'s (2025) work. The results in Table 4 show that SOaP yields higher performance. We argue that the improvement stems from SOaP correcting structural noise, while RASA corrects only positional noise. We also note that RASA requires training 9 linear layers of dimension $D \times 2$ ($D = 768$ for ViT-B), while SOaP is **simpler**, requires **no training**, and can be **attached to any model** as a single $D \times D$ linear head.

### 5 DISCUSSION

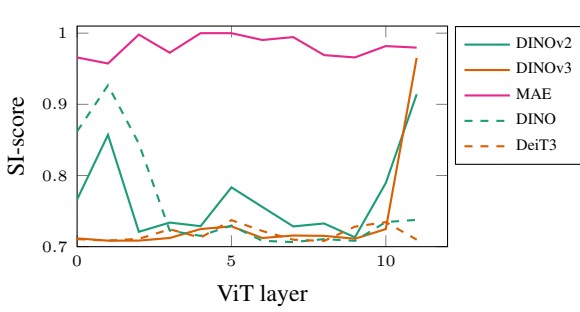
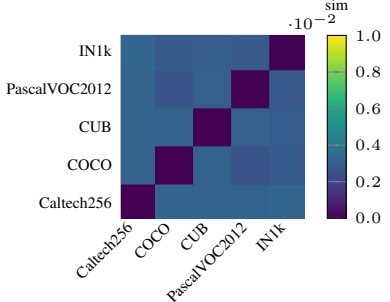

**Figure 6:** Maximum semantic invariance score for MIM models (solid line) and non-MIM models (dashed line). Critically, MIM models show high SI-scores in the last layers. This can be explained by the MIM objective encouraging positional information in the patch embeddings of deeper layers.

**Figure 7:** Cosine distance between SI-scores for DINOv2 using various datasets for semantically informative images. The low values in $(.0025, .0032)$ indicate that the SI-score is consistent across datasets.

Our analysis indicates a tendency for MIM to "cheat" to perform the matching of masked tokens by dedicating capacity to purely structural information. Furthermore, Figure 4 shows that this happens in both latent- and reconstructive MIM. The results are not totally unexpected; the reconstruction objective requires positional information to perform the task to some extent.

**Table 4:** Comparing SOaP with RASA on Franca ViT-B/14 for zero-shot salient segmentation on ECSSD (Yan et al., 2013) and kNN classification on ImageNet (Deng et al., 2009).

| Method | ECSSD (Sal. Seg) | | | IN1k (kNN cls.) | |
|---|---|---|---|---|---|
| | max $F_\beta$ | IoU | Acc. | Acc@1 | Acc@5 |
| Franca | 71.615 | 64.899 | 83.982 | 64.920 | 85.872 |
| Franca + RASA | 68.220 | 68.220 | 85.935 | 64.890 | 85.886 |
| **Franca + SOaP** | **84.176** | **76.985** | **91.514** | **65.084** | **86.006** |

However, we note that latent MIM models exhibit this property with or without additive positional encoding, such as for DINOv3. This means that the model learns to dedicate capacity to non-semantic structural information. This makes sense if we consider how much this actually helps the objective; by being able to correctly select the patch position in the image, the search is reduced by a ratio of $H_z \times W_z$. For a ViT-B/14 model, this results in a reduction of $\times 256$, significantly improving the loss of the model. While MIM improves performance on dense objectives, it does so at a *cost*. The importance and severity of this structural noise is corroborated by several works (Wang et al., 2024; Darcet et al., 2025; Yang et al., 2024). Our method shows that *the problem is pervasive for MIM models, and does not meaningfully occur in non-MIM models*.

**Limitations and Further Work.** We restrict evaluation to raw patch embeddings; both kNN and TokenCut operate directly on the representations without additional projections or heads. This is a conscious choice, as further transformations would confound the effect of SoAP with that of the evaluation model itself, which is beyond the scope of the current work. An avenue for future work is to study how SoAP interacts with more elaborate evaluation protocols.

# 6 CONCLUSION

We show that masked image modeling (MIM) objectives bias vision transformers toward encoding structural noise, at the expense of semantic content. This suggest that MIM learning signals are solved by short-cutting the intended objective of learning better local representation, and while this helps solve the pretext task, it reduces zero-shot generalization and semantic fidelity. To diagnose and address this issue, we introduce a Semantic Invariance Score and the lightweight, post-hoc denoising method SOaP, which consistently suppress structural noise and improve downstream performance. Our findings highlight a fundamental trade-off in MIM and offer practical tools for building more semantically robust self-supervised models.

## ETHICS STATEMENT

No ethical approval was needed as no human subjects were involved. All authors fully support the content and findings.

## REPRODUCIBILITY STATEMENT

We ensured reproducibility with publicly available datasets and standard models. Links to datasets, code, and configurations will be provided upon camera-ready submission.

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

**Figure C.1:** Figure illustrating the relation of soft responses of DINOv2 to $\text{PCA}_1$ in Figure 3 (left), compared with the multinomial distribution of activations $P_1$ from Figure 4 and Section 3.3 (right). The responses are binarized, and the probability distribution $P_1$ is computed as a multinomial distribution over all tokens in the image.

## A   DECLARATION OF USE OF LARGE LANGUAGE MODELS

During the preparation of this work, the author(s) used LLMs for grammar and spelling, as well as paraphrasing and rewording. After using this service, the author(s) reviewed and edited the content as needed and take(s) full responsibility for the publication's content.

## B   SELF-SUPERVISED OBJECTIVES

**Contrastive Learning.**   In this work, we focus on the contrastive self-distillation objective as presented in Caron et al. (2021). Given an image $x$, two random augmentations are applied yielding the views $u, v$. The views are sent through a teacher-student framework giving $p = f_\theta(v) \in \mathbb{R}^d$ and $q = f_{\hat\theta}(u) \in \mathbb{R}^d$, where $\theta$ and $\hat\theta$ denote the teacher and student weights, respectively. The loss minimizes the cross-entropy

$$\mathcal{L}_{\text{CLS}} = -q^\top \log p. \tag{B.1}$$

Importantly, this loss is applied on the global representations, given by the `CLS` token in ViTs. The teacher and student share the same architecture, comprising a backbone and a projection head for the global representation. The teacher is updated as an exponential moving average of the student.

**Masked Image Modeling.**   The core idea of the MIM objective is to reconstruct masked parts of an image when given visible parts as context. While the masking strategy varies between the MIM-based methods, the general setup can be summarized as follows. The input $x$ is obscured by a random mask $m$ and passed through an encoder to give a context $z = f(m(x))$ with which to make a prediction $\hat{s} = g(z)$ about the unmasked image $x$. The prediction can be made in the pixel space or in the latent space where the target $s$ is given by passing $x$ through the encoder. The loss is

$$\mathcal{L}_{\text{MIM}} = \ell(\hat{s}, s) \tag{B.2}$$

where $\ell$ is the mean squared error (He et al., 2022), euclidean distance (Assran et al., 2023) or negative cross entropy (Oquab et al., 2024; Siméoni et al., 2025; Zhou et al., 2022; Darcet et al., 2025). Notably, several frameworks (Oquab et al., 2024; Siméoni et al., 2025; Zhou et al., 2022) employ a combination of contrastive loss over the global `CLS`-tokens in a multi-view setup with a local MIM-based loss over masked patch tokens.

## C   DETAILS ON ACTIVATION AND DISTRIBUTIONS

In this section, we elaborate on the details of responses, activations, and distributions. As explained in the main text, we compute responses for all images in the dataset for each principal component. We binarize the activations, and compute the token-wise empirical distributions as individual Bernoulli distributions for each token in the image, $P_{d,n}$ from Section 3.3. As an ensemble, this can be taken as $P_d \sim \text{Multinomial}(2, N)$, forming a full distribution over the image.

For the scores, we choose the Dice-Sørensen, since it penalizes distributional uncertainty better than a pure divergence measure. Technically, if $P_{d,n} = Q_{d,n} = 0.5$, a proper divergence such as Jensen-

Shannon requires that the distributions are equal, e.g. $D_{\mathrm{JS}}(P_{d,n}, Q_{d,n}) = 1$, yet in our case it represents an uncertain activation. In contrast, Dice-Sørensen yields a score of $\mathrm{SI}(P_{d,n}, Q_{d,n}) = 1/2$ for the same example. For our purpose, this correctly identifies a form of uncertainty in contrasting responses between the real data and the non-semantic synthetic data, which yields clear principal directions of invariance to semantic content. Figure Figure C.1 illustrates the relation between responses and empirical estimates of the distribution.

### C.1 Form of Dice-Sørensen Coefficient

When we use the Dice-Sørensen coefficient in Section 3.3, we use it over multinomial distributions $P_d, Q_d$. To clarify, we consider the inner products and norms as being taken over the *support set*, not the multivariate dimensions.

Let $P_d, Q_d \in \mathbb{R}^N$ and let $1 \in \mathbb{R}^N$. Define the augmented vectors

$$\tilde{P}_d = [P_d, 1 - P_d], \qquad \tilde{Q}_d = [Q_d, 1 - Q_d] \in \mathbb{R}^{2N}.$$

Then

$$s_d = \mathrm{SI}(P_d, Q_d) = 2 \frac{\langle \tilde{P}_d, \tilde{Q}_d \rangle}{\|\tilde{P}_d\|_2 + \|\tilde{Q}_d\|_2}, \tag{C.1}$$

where $\langle \cdot, \cdot \rangle$ and $\| \cdot \|_2$ denotes the Euclidean inner product and norm.

## D Generating synthetic images

Let $\Omega = \{1, \dots, H\} \times \{1, \dots, W\}$ and $X \in \mathbb{R}^{C \times H \times W}$. For each image, draw mixture weights

$$\mathbf{w} = (w_1, w_2, w_3) \sim \mathrm{Dir}(\alpha_1, \alpha_2, \alpha_3). \tag{D.1}$$

We generate three components independently:

1. **Pink Noise**: $X_{\mathrm{pink}}$ is zero-mean with isotropic power spectrum

   $$\mathbb{E}\big[|\mathcal{F}\{X_{\mathrm{pink}}\}(\xi)|^2\big] \propto \|\xi\|^{-\beta}, \qquad \xi \in \mathbb{Z}^2 \setminus \{0\},$$

   with $\beta \approx 2$, following the power-law slope typical of natural images (Simoncelli & Olshausen, 2001).

2. **Modulated White Noise**: $X_{\mathrm{white}} = M \odot W$, where $W \overset{\mathrm{i.i.d.}}{\sim} \mathcal{N}(0, 1)$ and the nonnegative modulation $M = g(P)$ is a smooth field obtained from a pink process $P$ (as above) and a bounded mapping $g$ that sets the local standard deviation. This yields a heteroscedastic Gaussian field with variance $\sigma^2(x) = M(x)^2$ and long-range variance correlations.

3. **Gradient Field**: $X_{\mathrm{grad}}$ is a random low-degree polynomial—or equivalently, a very low-pass random field—concentrating energy near $\xi = 0$.

Synthesized images are then given by the convex mixture

$$X = w_1 X_{\mathrm{white}} + w_2 X_{\mathrm{pink}} + w_3 X_{\mathrm{grad}}. \tag{D.2}$$

Assuming zero mean and independence between components, the expected power spectrum of $X$ is

$$S_X(\xi) = \mathbb{E}\big[|\mathcal{F}\{X\}(\xi)|^2\big] = \sum_{i=1}^{3} \mathbb{E}[w_i^2] S_{X_i}(\xi), \tag{D.3}$$

i.e., a convex combination of the components' spectra.

Simoncelli & Olshausen (2001) show that natural images exhibit approximate scale invariance with a $1/\|\xi\|^\beta$ law in the power spectrum (amplitude $\sim 1/\|\xi\|^{\beta/2}$), together with large-scale illumination/contrast fluctuations. In the construction above, $X_{\mathrm{pink}}$ directly imposes the $1/\|\xi\|^\beta$ decay,

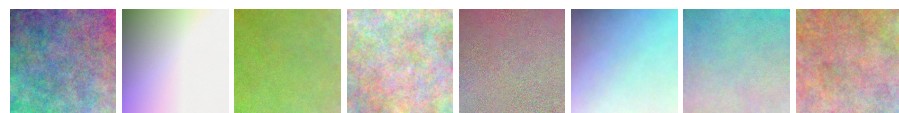

**Figure D.1:** Examples of generated synthetic non-semantic images.

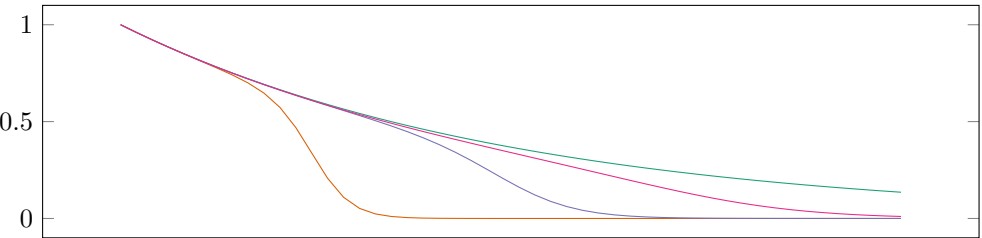

**Figure D.2:** Plot showing the effect of filtering with the Fermi window (7) on an exponentially decaying function with various choices of $\mu, \tau$.

giving second-order statistics aligned with natural image ensembles. $X_{\mathrm{grad}}$ injects additional low-frequency near-DC energy, modeling global trends and illumination. $X_{\mathrm{white}}$ introduces spatially varying contrast via a pink variance field, capturing long-range correlations of local variance found in natural scenes.

Thus $S_X(\xi)$ inherits a natural-image-like spectrum: a power-law falloff dominated by $X_{\mathrm{pink}}$, boosted near $\xi = 0$ by $X_{\mathrm{grad}}$, and with heteroscedasticity from $X_{\mathrm{mw}}$. We illustrate examples of synthetic images in Figure D.1.

## E    EVALUATION DETAILS

Throughout the paper, we provide several evaluations and experiments. In this section, we exposition some of the details for each evaluation method.

### E.1    TOKENCUT

We use the official TokenCut (Wang et al., 2023) implementation with their graph cut segmentation algorithm for the patch embeddings and the bilateral solver for edge-aware post-processing to refine the segmentations up to original image size. TokenCut employs the key features of the last attention layer as the input features to the graph cut algorithm. Contrary to the author's suggestion, we find that using the final output features yields better results for all models except MAE. Our reported results are thus for the out features for all models in our study, except MAE, for which we use the key features. We set $\tau = 0.3$ for all models; table E.1 shows this setting yield better results for DINOv2 and CAPI. Otherwise, we follow original implementation.

**Selecting foreground partition using salient principal components.**    We also observe that some principal components have a strong center bias in the activations. This can be partially explained by object center bias in the training data. Comparing with the activation maps of synthetic input data in Figure 4 shows that in several cases the center bias is not present. This indicates that these components are responding to relative saliency or instance level correlations for each of the local patches, rather than an encoded positional center bias in the model.

Given a principal component $v_d$ with salient activations, we can effectively determine which patches are likely to be part of the foreground or class level object. We find that we can improve zero-shot salient segmentation with TokenCut by selecting the foreground partition based on the patch with the highest response $\langle v_d, z \rangle$. In contrast, TokenCut selects the patch with maximum absolute value in its feature vector. The results in Table E.2 show out-of-the-box improvement across the board, where we use the strongest salient principal component to guide foreground selection for each model. We show results for all MIM models in our study, except I-JEPA which did not have a good salient principal component. We observed low performance on salient segmentation for all our experiments with I-JEPA, which suggests that the patch embeddings are not informative for this task.

**Table E.1:** Ablation over the TokenCut parameter $\tau$ on ECSSD (Yan et al., 2013) for DINOv2 and CAPI.

| | DINOv2 ViT-B/16 | | | CAPI ViT-L/14 | | |
|---|---|---|---|---|---|---|
| $\tau$ | max $F_\beta$ | IoU | Acc. | max $F_\beta$ | IoU | Acc. |
| 0.2 | 79.037 | 70.033 | 87.720 | 64.927 | 53.906 | 78.027 |
| 0.3 | 79.803 | 71.751 | 87.953 | 72.456 | 66.083 | 84.334 |
| 0.4 | 79.177 | 71.708 | 87.655 | 70.604 | 64.627 | 84.139 |

**Table E.2:** Using salient principal components to select foreground partition in TokenCut. Results are shows for corrected embeddings after cleaning with SOaP.

| | | Max. abs. val. | | | Max. Sal. PC reponse | | |
|---|---|---|---|---|---|---|---|
| Model | Arch. | max $F_\beta$ | IoU | Acc. | max $F_\beta$ | IoU | Acc. |
| DINOv2 | ViT-B/14 | 71.461 | 64.129 | 83.292 | 80.633 | 72.559 | 88.687 |
| DINOv3 | ViT-B/16 | 33.360 | 25.111 | 46.521 | 39.342 | 31.982 | 58.351 |
| iBOT | ViT-B/16 | 64.432 | 57.978 | 80.093 | 66.557 | 60.167 | 78.340 |
| MAE | ViT-B/16 | 80.525 | 70.705 | 90.506 | 82.094 | 72.118 | 91.444 |
| CAPI | ViT-L/14 | 74.506 | 68.045 | 86.3234 | 85.219 | 78.084 | 92.600 |
| Franca | ViT-B/14 | 77.660 | 70.830 | 87.577 | 84.176 | 76.985 | 91.514 |

## E.2 KNN CLASSIFICATION

We perform kNN classification on ImageNet (Deng et al., 2009) by average pooling the patch embeddings, and matching the validation embeddings to the $k$ nearest embeddings from the training set. We follow the kNN evaluation script by Caron et al. (2021), and set $k = 20$ number of neighbors and 0.07 temperature for the voting coefficient.

