# OpenReview forum: "Does MIM cheat? Exploring Semantic Invariance in Self-Supervised ViT Representations"
_ICLR.cc/2026/Conference — ICLR 2026 Conference Withdrawn Submission_

### Official Review · Reviewer_QSBR · 2025-10-21

**Soundness:** 3
**Presentation:** 2
**Contribution:** 3
**Rating:** 6
**Confidence:** 4

**Summary:**

The authors demonstrate that Masked Image Modeling (MIM) objectives tend to bias learned representations toward non-semantic cues. To address this, they introduce a model-agnostic counterfactual score that quantifies semantic invariance, enabling a clearer characterization of the tradeoff between semantic content and structural noise in Vision Transformer (ViT) embeddings. Building on this insight, they propose Semantic Orthogonal Projection (SOaP), a post-hoc method that enhances performance on downstream tasks across a range of state-of-the-art MIM-based models.

**Strengths:**

* The authors conduct an in-depth analysis of structural noise in Vision Transformer (ViT) token representations across state-of-the-art Masked Image Modeling (MIM) and contrastive learning models.

* Their evaluation spans a diverse set of models, enhancing the generality and robustness of their findings.

**Weaknesses:**

In Section 3.2, the authors assume that the patch embedding can be approximated as a linear projection. However, they do not provide sufficient justification for the validity of this approximation, leaving its soundness in practical settings unclear.

**Questions:**

* The rationale centers on using PCD to extract semantic information for guiding MIM. However, why is a relatively simple method like PCD capable of effectively guiding more sophisticated MIM models?

* In Figure 1, why is structural noise observed only in MIM models, while non-MIM models appear unaffected? What underlying factors contribute to this discrepancy?

* Could you clarify how to interpret Figure 3? Specifically, what are the key differences between the first and second rows?

* In the experimental setup described in Section 4.2, why is the linear probe metric omitted from the evaluation? Including it could offer additional insight into representation quality.

---

### Official Review · Reviewer_Daow · 2025-10-22

**Soundness:** 1
**Presentation:** 2
**Contribution:** 2
**Rating:** 2
**Confidence:** 4

**Summary:**

This paper considers the representations of image patches made by Masked Image Models (MIM).
It describes how the representations of MIMs contain both semantic information, e.g. information about which categories the content of the image falls into,
and non-semantic information, e.g. position of the image patch in relation to the full image.

The paper introduces a semantic invariance (SI) score, which can be used to measure how much non-semantic information is captured in certain directions of
the representations. It also proposes a method called Semantic Orthogonal Projection (SOaP) for supressing the non-semantic directions of the representations.
SOaP is related to the RASA method from [1] in the sense that this paper finds non-semantic directions using PCA and removes them using Gram-Schmidt,
while RASA finds the non-informative directions using a small trained model and then also removes them using Gram-Schmidt.

They run experiments on how much SOaP improves performance on the two downstream tasks salient segmentation (on 3 datasets) and
k-nearest neighbour (kNN) classification (on 1 dataset) for six MIM trained and two non-MIM trained models.
They find some improvement for the segmentation task for MIM models and no real change for the kNN task.
They also compare using SOaP with using RASA from [1] on a single model on a single dataset for each task. They find improvement on the segmentation
task, but no change on the kNN classification.


[1] Shashanka Venkataramanan, Valentinos Pariza, Mohammadreza Salehi, Lukas Knobel, Spyros Gidaris,
Elias Ramzi, Andrei Bursuc, and Yuki M. Asano. Franca: Nested matryoshka clustering
for scalable visual representation learning. arXiv preprint arXiv:2507.14137, 2025.

**Strengths:**

**S1:** Figuring out how to extract the desired properties from representations for use in downstream tasks is very relevant.
Especially with the large number of pre-trained models now available.

**Weaknesses:**

**W1:** The mathematical part of the paper is confusing, since notation is used without being introduced and many choices are not well-motivated.
See questions **Q1, Q2, Q4-Q7**.


**W2:** A lot of information is lacking with respect to the experimental setup.
For example:
- It is not explained how the hyperparameters for equation 7 are chosen.
- It does not say what $\beta$ is used in their $F_{\beta}$ measure.
- There are no descriptions of the datasets used.
- For segmentation, it says they use TokenCut (Line 362), but there is no proper description of how this method works (there are two sentences in appendix E.1 which is not referenced).


**W3:** The results are not convincing: Only two downstream tasks are tested and in only one of them does SOaP show any real effect.
In the segmentation case, it seems there is some effect (table 2) since results improve for all models which somehow include MIM.
However, the size of the effect is not possible to estimate from these results, especially since there is only one
seed per model type. Table 3 about kNN classification does not show enough change to make any conclusion.


**W4:** The paper is very unclear in several places: Sometimes references are used with no clear reason why the reference is relevant
to this part of the text. Captions for figures and tables are also lacking in explanations. See questions **Q3, Q8-Q10**.

**Questions:**

This submission is very unclear both with respect to explaining the theoretical foundation and how one might reproduce the results.
I therefore recommend rejection.


**Questions:**


**Q1:** Line 160: What is $\mathcal{Z}$ here? Do you mean $z\in \mathcal{Z} \subseteq \mathbb{R}^D$, where $D$ is the dimension of the embeddings?


**Q2:** Line 161: Are you saying that you use Welford's online algorithm to calculate the eigendecomposition? Welford's is an algorithm for
calculating the variance of an entire dataset using batches. It does not give you the eigenvectors.


**Q3:** Figure 3: How did you choose which components to pick? Is this related to figure 5?
In the bottom row for DINOv2, iBOT and MAE, it looks like there is a rough segmentation going on which is not exclusively top/bottom based.
Do you have a comment on this?


**Q4:** Equation 2: Should the inequality with $\tau$ not be strict? If seems strange to say there is an "activation" when the projection of
the representation onto the eigenvector is zero. Especially when you in line 244 say that similar activation vectors for inputs should
mean that they have similar behaviour with respect to the eigenvector.


**Q5:** Line 192: Why not just set $\tau = 0$ here? Why define it as a dot-product?


**Q6:** Line 193: You have not introduced $B$. I assume $B = \vert \Omega\vert$?


**Q7:** Line 223: "increasing the coefficient $\theta_{\rho}$ at the expense of the semantic coefficient $\theta_{\phi}$."
	Why do you say there is a trade-off? In your equation 3, $\theta_{\rho}$ and $\theta_{\phi}$ are not linked, that is,
	$\theta_{\rho}$ can go up without $\theta_{\phi}$ going down.
	Do you mean that if $\theta_{\rho}$ is large compared to $\theta_{\phi}$ it is more difficult to pick out
	$\theta_{\phi}$ from the representation?


**Q8:** Line 231: You reference [1] as a source for semantic invariance, but your definition of semantic invariance seems
to differ from the one in [1]. If you claim they are the same, could you explain how?


**Q9:** Line 264: You reference [2] after "Gram–Schmidt based projection", why is this? I would guess it is because they
also use Gram–Schmidt in [2] to remove the non-informative directions? However, this is not clear in the text.


**Q10:** Equation 7: How is this function related to a Fermi window? How do you choose $\mu$ and $\tau$ in your experiments?
Also, you already used $\tau$ for your threshold in equation 2, so you should find a different name for this new hyperparameter.





**Additional Feedback (no effect on recommendation):**

**F1:** Saying that MIM "cheats" when it puts information about position into its embeddings, I find slightly misleading,
since position is very important information when considering the training objective.
As such, I am not fond of the "Does MIM cheat?" part of the title.


**F2:** Remember to always introduce an abbreviation before you use it.
For example in the first line of the abstract write "masked image modeling (MIM)" instead of just MIM.


**F3:** Figure 1 + 2: Please be more clear about the objective in these figures. In figure 1: Is the blue patch supposed to cover only the pyramids?
Or the entire foreground, pyramids + sand? Does IJEPA simply not mark any pixels in either case?
In figure 2: Is the objective to separate foreground and background or to mark the object for classification?
I am assuming that marking the chicks is the objective, since you mark this as correct in the figure, but please be more clear about it in the caption.
When I read this, I did not know what TokenCut was. Make sure to add a reference in the caption, also to your appendix E.1.


**F4:** Table 2 and 4: Make sure to write in the caption or at least somewhere in the paper what you are measuring.
For example, what is max $F_{\beta}$?




Typos:

Line 088: "improved representations for downstream-cf.", I think it should say "downstream tasks".

Line 096: "and are able to" -> "and is able to"

Line 102: "has a extensive" -> "has an extensive"

Line 147: "sheds insights to decompose" -> "sheds light on how to decompose"

Line 257: "With SI score (5)" -> "With the SI score (5)"

Line 263: "is then defined following Gram–Schmidt based" -> "is then defined following a Gram–Schmidt based"


References:



[1] Bo Yuan, Danpei Zhao, and Zhenwei Shi. Learning at a glance: Towards interpretable data-limited
continual semantic segmentation via semantic-invariance modelling. IEEE Transactions on Pattern
Analysis and Machine Intelligence, 46(12):7909–7923, 2024.

[2] Shashanka Venkataramanan, Valentinos Pariza, Mohammadreza Salehi, Lukas Knobel, Spyros Gidaris,
Elias Ramzi, Andrei Bursuc, and Yuki M. Asano. Franca: Nested matryoshka clustering
for scalable visual representation learning. arXiv preprint arXiv:2507.14137, 2025.

---

### Official Review · Reviewer_VW5d · 2025-10-31

**Soundness:** 2
**Presentation:** 3
**Contribution:** 2
**Rating:** 4
**Confidence:** 3

**Summary:**

This paper investigates the adverse effect of MIM objectives in driving ViT to learn non-semantic “structural” information. In particular, the authors (1) design a model-agnostic semantic invariance (SI) score to quantify this tradeoff and (2) propose Semantic Orthogonal Projection (SOaP), a post-hoc projector based on Gram–Schmidt with a Fermi window weighting that helps remove non-semantic information from the patch embeddings. The experiments on several ViT-based MIM models show that SOaP improves the performance in downstream tasks (TokenCut salient segmentation & kNN classification, with less improvement for the latter). The analysis serves as a step forward to quantify this important trade-off in MIM training and points out a gap in learning semantically relevant information.

**Strengths:**

* The problem is very relevant to several active research areas (e.g. SSL, spurious correlations, inductive bias in OOD generalization). It is framed and motivated clearly. Subsequently, the model-agnostic, label-free SI score is derived in an intuitive and conceptually clean way.
* The experiments show empirical performance gain and cover popular SSL families; the proposed methods are fairly straightforward and easy to implement.

**Weaknesses:**

* **Oversimplifying Assumption**: The theoretical justification relies on the assumption that we can linearly decompose the embedding into semantic and structural information. However, if structural and semantic factors are nonlinearly entangled, the boundary becomes “blurry”, and suppressing top PCs could also remove useful features (see next part). For simplicity in theory, this assumption is reasonable to me, but the central claim “MIM cheats by encoding structure” would benefit from more complementary studies.

* **Degradation in non-MIM models**: From Table 2, we can see that while SOaP improves the performance on saliency segmentation for MIM models. For DINO, it also degrades all the metrics by a large gap, which suggests that the method can also remove relevant semantic information about the embeddings. Also the improvements on kNN classification seem very incremental to me.

* **Limited Experiments**: The evaluation covers TokenCut saliency and kNN as the main downstream tasks. The method would become more convincing if it leads to improvement on other tasks like linear probing. The authors did partially justify this in the Limitations section that many tasks require more than raw embeddings, but the potential existence of confounders limits the generalizability of this work. Would a preliminary investigation into confounding be possible?

**Questions:**

Most questions are raised in the Weakness section. Additionally,

1. The non-semantic distribution is a specific type of synthetic data (detailed in the appendix). Have you tried other counterfactual designs?

2. How are the hyperparameters selected for the Fermi window? Are they purely empirical or do we have some heuristics?

3. Figure D.2 should be labeled more clearly.

I am happy to raise my score if the above concerns are addressed.

---

### Official Review · Reviewer_Hkga · 2025-10-31

**Soundness:** 3
**Presentation:** 3
**Contribution:** 3
**Rating:** 4
**Confidence:** 4

**Summary:**

This paper proposes a new measure of structural noise (a generalisation of positional noise) in ViTs and uses it to design a training-free post-hoc method to non-informative content from patch embeddings. The authors begin by observing that, as reported in previous studies, MIMs encode positional information, which hinders their performance on downstream tasks. Assuming that patch embeddings can be linearly decomposed into positional and semantic components, they perform a PCA on a set of patch embeddings and compute the (binary) activations one would obtain by multiplying the patch embeddings by the previously obtained PCs. After observing that spatial patterns emerged in MIMs (but not in other ViTs) when averaging these activations across images, the authors hypothesized that PCs representing structural information should be similarly activated in semantic and non-semantic images. They measured the similarity between activation distributions obtained from a semantically meaningful image and one devoid of semantic content (e.g., white noise) and used the obtained score to attribute a higher weight to non-semantic components and remove them from the patch embeddings.

**Strengths:**

- The paper is clear, easy to follow, and well-structured.
- I like the idea of a training-free post hoc method to remove structural noise from patch embeddings. It is simple, elegant, and seems to perform well.
- SOaP appears straightforward to apply and could be beneficial for any practitioner using ViT embeddings in downstream tasks. I could definitely see myself using it in my research.
- SOaP outperforms existing methods (RASA) and generalises well to various ViT models and datasets.

**Weaknesses:**

While I like the overall idea of the paper, my main concern is that some design choices seem unnecessarily complex, and their need is not entirely justified in the paper (see questions 1-4). For example, if a generalisation of positional noise is required, did the authors identify other types of non-positional structural noise?  Yes, SOaP outperformed RASA, but how can we be sure that it is because RASA only captures positional noise? Similarly in 3.3, I didn't understand why specifically choosing Dice-Sørensen until I read appendix C and I am still not sure why we need Fermi window at all for 3.4 as I didn't see any results showing that SOaP didn't work well using $s_d$ directly.

Other comments:
- In 3.1, the mathematical notation in 3.1 could be improved. There are some inconsistencies between upper and underscript $b$ (see minor comments).
- Figure 5 and D2 could be improved (see minor comments)
- There are a few stray typos (see minor comments)

**Questions:**

1. I understand that positional noise is a generalisation of structural noise, but could the authors explain why it is needed? Can we definitively attribute the better performances of SOaP over RASA are imputable to this generalisation? Did the authors observe other types of structural noise in MIMs? If so, could they describe them?
2. Could the authors put the part about Dice-Sørensen metric penalisation of uncertainty from App. C in 3.3? It was quite frustrating to have a justification for this so late in the paper. Also, in which case would there be such uncertainty in the current setup?
3. Could the authors provide the results obtained by SOaP without Fermi window (e.g., using $s_d$ instead of $w_d$)? If the results are worse, it would justify the need for a more complex strategy. However, depending on how well that version of SOaP performs compared to RASA, it may invalidate the theory that RASA is underperforming because it only takes into account positional noise.
4. In 3.1., why did the authors choose $\nu = 0$ ? Was it because this led to better results? Setting $\nu = 0$ implies that we have $\tau=0$, correct? If so, and if $\nu \neq 0$ leads to poor results, why not directly use $\tau = 0$ instead of doing an additional inner product? It would be great to have these results in the appendix for completeness.
5. What is the impact of the number of samples $z$ used to compute the covariance matrix? Did the authors observe important changes in SOaP performances when varying this?
6. In C.1., the authors emphasise that they compute the norms and inner products of $s_d$ over the support set. How is that implemented in practice?

Minor comments
=============
- in 3.1 the authors used $z^{(b)}$ but $x_b$. This inconsistency hampers the reading, as one may think that these two $b$ indicate different things. I would suggest replacing $x_b$ by $x^{(b)}$ and changing $x^c \sim \mathcal{X}^c$ to $\tilde{x} \sim \tilde{\mathcal{X}}$ in Eq. 4 to make the notation more consistent.
- l.85 representaitons should be representations
- l. 246 $Q_{d,n}$ and  $Q_{d,n}$ should be  $P_{d,n}$ and  $Q_{d,n}$
- Fig. 5 is conmfusing because 5(a) lacks an x axis. Furthermore, these are not continuous data and they would be better represented by point plots.
- Could the authors provide a legend for the 2 y-axis and the x-axis of Fig. D2?

---

### Note · Authors · 2025-11-14

**Comment:**

We wish to withdraw our submission, as we believe the paper requires additional refinement before publication.

Best regards,
The authors

**Withdrawal Confirmation:**

I have read and agree with the venue's withdrawal policy on behalf of myself and my co-authors.